# Current Methods of Newborn Screening Follow-Up for Sickle Cell Disease Are Highly Variable and without Quality Assurance: Results from the ENHANCE Study

**DOI:** 10.3390/ijns10010022

**Published:** 2024-03-08

**Authors:** Najibah Galadanci, Shannon Phillips, Alyssa Schlenz, Nataliya Ivankova, Julie Kanter

**Affiliations:** 1Division of Hematology and Oncology, Department of Medicine, Heersink School of Medicine, The University of Alabama at Birmingham, Birmingham, AL 35233, USA; ngaladanci@uabmc.edu; 2Department of Pediatrics, College of Nursing, Medical University of South Carolina, Charleston, SC 29425, USA; phillipss@musc.edu; 3Department of Pediatrics, University of Colorado School of Medicine, Aurora, CO 80045, USA; alyssa.schlenz@childrenscolorado.org; 4School of Health Professionals, Health Services Administration, The University of Alabama at Birmingham, Birmingham, AL 35233, USA; nivankov@uab.edu

**Keywords:** newborn screening, sickle cell disease, follow up, qualitative interviews

## Abstract

Newborn screening (NBS) for sickle cell disease (SCD) has significantly improved childhood survival but there are still gaps resulting in delayed care for affected infants. As a state-run program, there are no national quality assurance programs to ensure each state achieves consistent, reliable outcomes. We performed this qualitative study of NBS follow-up practices to better evaluate and understand the multi-level, state-specific processes of how each state’s public health department delivers the NBS results to families, how/if they ensure affected infants are seen quickly by sickle cell specialists, and to determine the close-out processes used in each state. This project used semi-structured interviews conducted with 29 participants across eight states to explore these NBS follow-up processes in each state. Participants included SCD providers, NBS coordinators, or personnel associated with state health departments and community-based SCD organizations (CBO). Our results show significant state-dependent variations in the NBS processes of information delivery and patient management. Specifically, programs differed in how they communicated results to affected families and which other organizations were informed of the diagnosis. There was also state-based (and intrastate) variation in who should assume responsibility for ensuring that infants receive confirmatory testing and are promptly started on penicillin prophylaxis. Case closure was also highly variable and poorly validated. Our results also yielded identifiable challenges and facilitators to NBS which were highly variable by state but potentially addressable in the future. This information suggests opportunities for systematic improvement in NBS follow-up processes.

## 1. Introduction

Sickle cell disease (SCD) is one of the most common severe monogenic disorders with about 300,000 children born each year worldwide [1,2,3]. SCD affects 1 of every 396 Black newborns with at least 100,000 persons living with SCD in the United States (US) [1,4]. Due to the high early mortality in affected newborns, the US implemented a newborn screening (NBS) program for SCD in 1975 [5]. By 2006, all 50 states included SCD on their NBS panel, and it is the most common condition identified by the universal NBS program [6]. NBS for SCD is performed in order to ensure children are diagnosed early so they can be immediately referred to a comprehensive SCD center for initiation of effective treatment and education [7,8]. As a result of NBS and other early interventions (including penicillin prophylaxis, immunizations, and early education), over 95% of affected children with SCD grow into adulthood [9,10,11]. The introduction of the seven-valent pneumococcal conjugate vaccine also contributed to a drastic drop in mortality rate (68%) in children under 3 years with SCD [12]. Additional benefits of NBS include prompt clinical intervention for infection or splenic sequestration episodes, and early caregiver education about the signs and symptoms of illness in infancy and early childhood [13]. Compared to countries with lower resources where more than 50% of children die prior to age 5, over 95% of children with SCD reach adulthood in the United States [11,14,15].

Despite the clear success of the NBS program for children with SCD, there are critical gaps in our care delivery system. We previously published data from healthcare providers who described significant variation in both the initial reporting of NBS results to families and establishing care with the SCD clinic [16]. Many of the discrepancies are due to the structure of NBS programs. NBS programs are administered by state-specific departments of public health (DPH) that do not use standardized protocols for notifying families of the SCD diagnosis or ensuring appropriate or timely follow-up. Outside of the laboratory testing performed, there are no standard quality assurance processes in NBS for SCD [17,18]. Thus, despite the significant improvement in childhood survival for SCD, there are critical gaps in our current understanding of how the initial information/diagnosis translates to effective care delivery. At present, there is remarkable variability in how families are informed of the SCD diagnosis, to whom they are referred, and what initial information they receive.

Current data suggest improvement in SCD survival is due to the institution of universal NBS. However, the widespread and universal use of the protein-conjugated pneumococcal vaccine was initiated in the same decade, making it difficult to attribute the improved survival to NBS alone [19,20]. Current data suggest that the goals of the NBS program including the initiation of penicillin prophylaxis are not as robust as intended, with <30% of infants with SCD started or maintained on penicillin prophylaxis [21]. Similarly, several studies including our group’s DISPLACE (Dissemination and Implementation of Stroke Prevention-Looking at the Care Environment) study have demonstrated the poor implementation of stroke screening for young children with SCD as well [22,23,24,25,26]. These findings are some of several examples suggesting that there is insufficient implementation of NBS follow-up and ongoing care engagement [21]. It is also unclear whether all children with SCD are seen by a SCD specialist within the first 3 months of life, as recommended by the National Heart Lung and Blood Institute (NHLBI) expert guidelines [27]. To better understand these gaps, we have initiated a long-term project, entitled ENHANCE (ENHAncing Newborn screening Continuing Engagement). The goal of this multi-pronged project is to evaluate current NBS follow-up practices and their effect on long-term SCD care engagement. ENHANCE is a multi-level, multi-center project that includes both qualitative and quantitative assessments in order to develop an intervention to improve long-term care engagement for SCD. This paper describes the initial, qualitative part of the ENHANCE project. The purpose of this qualitative study was to explore NBS programs’ follow-up practices for SCD at multiple levels to obtain an in-depth understanding of systems of delivery of NBS reports to families, timing of clinic follow up, close out processes for SCD cases, and to identify barriers and facilitators to NBS follow-up.

## 2. Methods

### 2.1. Design, Sample, and Setting

We used a qualitative descriptive approach [28] to explore healthcare providers’ and stakeholders’ experiences with NBS for SCD. Participants were solicited from SCD centers, sickle cell community-based organizations (CBO), and DPH in 11 states across the US, including Alabama (AL), California (CA), Georgia (GA), Arkansas (AK), South Carolina (SC), Tennessee (TN), Michigan (MI), Mississippi (MS), Missouri (MO), New York (NY), and Louisiana (LA). Providers were selected if they provided healthcare to infants and young children with SCD in one of the target states. Stakeholders included employees from NBS programs and/or state departments of public health (DPH) SCD providers and/or individuals working with SCD-focused community-based organizations (CBOs) in one of the target states. Providers and stakeholders who had worked or volunteered for less than 6 months at the relevant organization were excluded for lack of experiential knowledge. Purposive and snowball sampling was used to identify potential participants with experience with NBS through professional networks and asking for additional contacts from participants at the end of each interview. Participants were approached by N.G. or J.K. via email and asked to volunteer for the study. A total of 34 representatives were approached via email and/or phone contact in the 11 states. A total of 33 out of 34 responded to the initial email. Out of the 33, 29 responded and agreed to our request to schedule an interview date. Of the 6 potential participants who refused participation, none provided reasons for refusal. Our estimated sample size was up to 20 interviews with providers and 20 with stakeholders; sample size was determined by data saturation, or the point at which new information was no longer forthcoming during interviews. 

### 2.2. Data Collection

Ethical approval to conduct the study was obtained from the University of Alabama at Birmingham Institutional Review Board (IRB) prior to data collection. Semi-structured interview guides were developed by S.P. and N.G. to explore variation in NBS processes and barriers and facilitators to NBS and included open-ended questions with probes. Verbal consent was confirmed following the investigator reading an introductory statement. Interviews were conducted by S.P., a female, PhD-prepared nurse scientist with extensive training and experience in qualitative research. The interviewer had a prior professional relationship with some, but not all, participants. For those with whom the interviewer did not have a prior relationship, S.P. introduced herself, her background, and her role in the study prior to initiating interviews. 

Demographic data were collected from each interview participant and included personal characteristics, such as age, race/ethnicity, gender, and professional characteristics, such as number of years interacting with individuals with SCD. Interviews were conducted virtually using Microsoft Teams© (Version 24004.1304.2655.7488) and audio recorded on a digital recorder. Participants were encouraged to be in a private location during interviews, and the interviewer conducted interviews in a private location, such as an office with a locked door. N.G. was present during interviews for observation and to record field notes. S.P. also collected field notes during and immediately following each interview. Interviews lasted 17–56 (mean 34.7) min. Audio files were sent to an external, Health Insurance Portability and Accountability Act (HIPAA)-compliant agency for transcription, and transcribed interviews were reviewed for quality and accuracy, and redacted for identifiable information by N.G. No repeat interviews were conducted, and transcripts were not returned to participants for correction or comment. Data saturation was reached at 19 interviews (12 stakeholders and 7 providers). The conduct and reporting of this study followed the COnsolidated criteria for REporting Qualitative research (COREQ) checklist [29]. 

### 2.3. Data Analysis

Directed content analysis was conducted using a hybrid deductive-inductive approach [30,31] and a constant comparison method [32]. An initial codebook was developed by S.P. and N.G. to define process stages and elucidate barriers and facilitators to NBS at each process stage. The development of the initial codebook was informed by the SCD care continuum model that we adopted from the HIV care continuum [33] and that outlines the steps that newborns with SCD go through from diagnosis to achieving and maintaining care engagement in an established SCD care (Figure 1). We assessed specific factors and outcomes, including barriers and facilitators relative to each stage (NBS process, communication of NBS results to families and providers, and referral and linkage to care) that will help us identify gaps and develop strategies to improve follow up and retention in care. 

The codebook was iteratively revised by S.P., N.G., and A.S. until consensus was reached with consistency in coding. Codes represented a hierarchical structure; the first two stages in the guiding process framework were explored, with themes within each process stage describing key characteristics, and barriers and facilitators within each stage. Transcripts were coded by N.G. and A.S.; S.P. double-coded 5 transcripts and discussed coding with N.G. Intercoder reliability and agreement were conducted using the method described by Campbell et al. [34]. Intercoder reliability was 71% and intercoder agreement was 94%. Coding was conducted using NVivo 14 plus, Microsoft Word, and Microsoft Excel. Following the completion of coding, S.P. double-coded 4 transcripts from the overall sample (20%) and determined coding remained consistent between N.G. and A.S. Participants did not provide feedback on findings.

## 3. Results

### 3.1. Participants

The demographic characteristics of the interview participants are presented in Table 1 and Table 2. A total of 29 stakeholders from 11 states were interviewed, including 8 health care providers and 21 individuals working either at the state DPH or CBO. The healthcare providers included pediatric hematologists, pediatric nurse practitioners, and pediatric hematology nurse practitioners with at least 5 years of experience working with individuals with SCD. The median age of the stakeholders 50 years, ~90% were female, and participants reported a mean of 11 years working experience with NBS. 

### 3.2. Process Stages and Themes

Two stages in the NBS process were explored: newborn screening and communication of results and referral and linkage to care (Figure 1). The newborn screening and communication of results process stage was defined as actions that take place among relevant parties from the time the blood sample/blood spot is collected until the referral to SCD care is in place. The referral and linkage to care process stage was defined as the actions that take place among relevant parties from the time the referral to an SCD clinic is put in place until the appointment is scheduled for the first SCD clinic visit. An overview of themes within each stage, including theme descriptions and illustrative quotes, is presented in Table 3. 

#### 3.2.1. Process Stage: Newborn Screening and Communication of Results

*Determining the Initial Diagnosis*. Across all states, blood spots for the NBS were taken on a card at the hospital or birthing center and the samples were transported to and processed in the state’s DPH. The timing of the receipt of results varied across states from 1–2 days to 1 week. States also varied in numbers of NBSs positive for SCD, which ranged from 60 to 200 new cases per year. Participants in two states commented on using specific databases for results, including one state that had results automated into a database.

Disorganization and confusion specific to receiving lab results from the NBS blood spot was a key barrier. An additional barrier described by participants was lack of adequate communication or miscommunication between responsible parties that led to missing infants with positive/abnormal screens and delayed linkage to care. Participants were especially concerned about the impact of these barriers on the health outcomes of infants with SCD. 

A key facilitator was having an electronic system, including electronic submission for ordering cards, notifying the responsible person that results are ready or that samples are received, and alert of abnormal results. The use of a courier service for transporting samples to the lab and partnering with neighboring states to learn from one another and improve the process were also facilitators. These facilitators were perceived to reduce human error and the likelihood of missing infants who screened positive. 

*Communicating Results to Providers and Families*. Interviewees explained how abnormal results were communicated to medical providers and, ultimately, to families of affected children. The information gathered showed that when results were communicated to families, providers, and/or the hematologist varied significantly (e.g., within 24–48 h of the diagnosis, same week sample was received, or longer). Across all states, physicians of record were notified of abnormal results, typically first by phone using contact information from the NBS card or by contacting neonatal intensive care unit (NICU) providers, if appropriate. Results were then faxed to the same physician in one state, with two states also supplying additional written educational materials about the results for the provider to review with the family. 

Different parties are involved in the delivery of results to families in different states and could include the DPH, CBOs, and/or primary care providers (PCPs). In four states, information was routinely communicated to both the pediatrician and the family by the DPH. The extent of information provided also varied from a simple notification of the result (with deferral to the pediatrician for more information; one state) to more extensive explanations of NBS and formal letters (two states) or face-to-face meetings in the home (one state). In other states, the DPH only intervened by directly contacting families if the PCP was not reachable (to verify contact information; three states), if the family did not have an identified PCP (one state), or if the PCP did not feel comfortable communicating results (one state). CBOs played a similar role in some states by obtaining contact information for the PCP, communicating results to the family if a PCP was not listed, or being responsible for providing the initial information on the NBS results. In one state, all NBS results were conveyed to two organizations that were contracted by the state to provide notification to parents of infants with abnormal results. In three states, a contract existed between the DPH and particular organizations that coordinated and managed the positive screenings. In another state, all abnormal results were sent to the SCD association that was responsible for reaching out to the parents and the PCPs and arranging for confirmatory testing. CBOs also tended to provide an ongoing educational role beyond the NBS results. Finally, in two states, the pediatric hematologist was responsible for communicating results to families and notified the DPH when the infant was seen.

The most frequently mentioned barrier to communication of the NBS result was difficulty reaching families (four states) due to inconsistencies in contact information, frequent changes to phone numbers, families changing addresses, families moving out of state, or disorganization by the PCP that led to lost contact information. In such instances, the main alternatives were to use the mailing address provided on the NBS card or contact the birth hospital to find more information. Various barriers related to communication between the DHS and PCPs and between PCPs and families were also described. Stakeholders noted challenges with communication if the wrong PCP was listed on the NBS card (three states) or if the infant was in the NICU (and thus was not yet established with the PCP of record; one state). The quality of communication from PCPs to families could also be barrier, such as PCPs lacking knowledge regarding interpreting the NBS result and SCD (three states) or providing incomplete information to families and assuming the DPH would provide education (one state). 

Additional barriers were related to negative family responses and fear of the results, such as families having emotional difficulty with the result due to prior negative experiences with SCD (two states), difficulty accepting the result (one state), confusion about the result due to lack of correct information about SCD (one state), or lack of knowledge about their own trait status (one state). In addition, communication could be difficult if language barriers were present (one state).

Facilitators identified for communicating NBS results included having back up methods for contacting families and obtaining correct information for pediatricians (eight states). Methods included the use of other databases or having individuals with access to other records (e.g., other state databases, immunization records, Medicaid/insurance records, hospital records), multiple contact numbers and/or family members who could be reached, and specific individuals (“case finders”) who helped to track down families or accurate information for the PCP. The latter group included follow-up coordinators, individuals in medical records departments, CBOs, and SCD providers/team members. In some states, this individual served as a communication link between PCPs and families. CBOs and SCD providers were more broadly described as facilitators for supporting the communication process overall (beyond just finding families). Another facilitator was having dedicated and well-trained staff at the birth hospital who ensure the demographics on the NBS card are filled out appropriately and that the contact information for the family and PCP were entered correctly (one state). One interviewee noted that families with prior knowledge of SCD had an easier time accepting the NBS result. As noted by this individual, families with other relatives with a child with severe SCD are usually scared of the diagnosis, while if the family member themselves already had an affected child and were familiar with the disease and healthcare experience, they accepted the diagnosis more readily.

#### 3.2.2. Process Stage: Referral and Linkage to Care

*Referring families and linking them with SCD care*. Processes were highly variable in terms of referral to and scheduling the first SCD clinic visit, but were predominantly driven by either the PCP, the SCD clinic, or the family. In one state, the PCP facilitated both referral and scheduling of the visit for the patient whereas in another state, only the PCP managed the referral. In two other states, the family was responsible for contacting the SCD clinic and scheduling the visit. Families in one state could choose the SCD clinic they wanted to attend, whereas in another state, families were referred to the center closest to them. In one state, the PCP or an SCD resource center facilitated referrals. Participants in five states mentioned the timing of the first SCD clinic visit. In three states, participants mentioned specific timelines for getting infants in for their first visit, which ranged from 2 to 6 weeks. Participants also mentioned having state funds to expedite the SCD clinic visit and having a CBO to follow infants until the confirmatory testing was completed and routine care established. Participants in three states discussed processes after the first visit was scheduled, which included emailing the infant/toddler’s multidisciplinary team after the visit was scheduled, emailing a reminder and clinic information to the family, and sending a letter to families with information about the clinic. 

Participants in four states described systems and databases as part of this process stage. In one state, there is a tracking system for infants to which the DPH, PCPs, and SCD providers have access, but the PCPs usually do not access the database to see needed referrals. Similarly, in another state, there is a database that SCD providers and PCPs can access to see if the visit with the PCP has taken place. In yet another state, the CBO has a registry for individuals with SCD. Finally, in one state, a participant described a developing process to work with health information exchange to improve tracking. 

Participants in three states described communication barriers including referrals not getting through to the providers, gaps in communication between the SCD clinic and the DPH, and families finding the information upsetting and having difficulties accepting the results. Participants in four states described challenges around the PCP or pediatrician. These challenges included the PCP not being a part of the NBS process, disorganized PCP practices, and either PCPs who did not want to refer the infant to a SCD clinic or families who only wanted to be seen by the PCP. Challenges with finding families to communicate results (noted previously) were also relevant to ensuring that referrals were made. Systems-level issues were mentioned by participants in five states as follows: complex systems for making appointments that led to delays in appointments (4 states), misinterpretation of confirmatory testing results (1 state), a lack of standards around reporting to the state (1 state), and a registration system in the state that was “clunky” and made it challenging for families to make appointments (1 state). 

Participants in four states described collaborations as facilitators, such as the SCD clinic and DPH having a good relationship, regular meetings between the SCD clinic and the DPH, and PCPs working with the SCD clinic to ensure infants saw a SCD provider as soon as possible. Across three states, participants described relationship-building with the family as facilitators, which included DPH personnel making face-to-face visits with families, accommodating family needs to get the infant into clinic, and reaching out to families when they are having trouble accepting the diagnosis. Participants in four states described services or actions as facilitators, which included providing education and information to families early in the process, having funding available to assist families in attending the first SCD visit, having patient advocates to assist families with needs, and having assistance with finding families to communicate results and schedule an appointment at the SCD clinic. Finally, participants in three states described databases or tracking systems as facilitators. 

*Confirming the SCD Diagnosis*. Participants in six states reported confirmatory testing was conducted exclusively or primarily at the SCD clinic but was sometimes conducted by a pediatrician or PCP, depending on circumstances such as delayed SCD clinic visit or rural residence. Three participants in two states reported confirmatory testing was conducted at either the SCD clinic, or if there are insurance barriers, at a CBO; in one of these states, families were encouraged to complete testing at the SCD clinic. Testing conducted at one CBO was grant-funded; similarly, grant funding in another state supported all samples being sent for DNA testing to confirm the genotype. Participants in five states mentioned reporting confirmatory testing results back to the state’s DPH. In one state, test results went directly to the DPH, who then disseminated results to the PCP and SCD clinic. A participant in another state described a state tracking system in which results were entered, and a participant in a different state mentioned sending a monthly report to the DPH that included confirmatory testing results. Participants in three states also described a timeframe within which confirmatory testing is conducted, all of whom stated a goal of within one to three months of receiving the positive screening result. 

*Case Closure at the level of DPH*. Several participants described the DPH would close a child’s case after the infant’s first visit at the SCD clinic (six states), although how this was communicated between the SCD clinic and the DPH differed. In addition, the information required by the DPH to close the case also varied but generally included dates when the confirmatory testing was conducted (and the result of the test), when PCN prophylaxis was started, and when the infant was seen at the SCD clinic. However, not all states required infants to attend the SCD clinic prior to closing the case. Participants in two states mentioned situations in which a case would remain open longer, which typically either occurred when there was an unidentified hemoglobin result or the family had not yet been found and contacted. 

## 4. Discussion

Results of this qualitative study highlight significant variation in NBS processes for SCD across the 11 states represented. The findings emphasize the lack of standardized protocols for NBS follow-up. Specifically, there are no protocols for how/when families are informed of a diagnosis, what initial education is given to affected families, how infants are linked to care, and whether there is assurance that children are seen by a SCD specialist as intended. Notably, there are no identified best practices or quality assurance in any of these areas. Unfortunately, these results further support our initial findings from healthcare providers who described variation in multiple areas of the NBS process [16].

Across all states, results of NBS were communicated to the primary care physician (PCP) on file for the newborn. However, the process of communicating results to the families varied significantly as noted above. These findings are validated by two other previous survey studies of NBS in the US that reported that NBS results are communicated to PCPs in all 50 states and that in more than half of these states, it is the responsibility of the PCPs to communicate the results to the families [17,35]. These findings are important because PCPs have varying levels of SCD-specific education to provide for families. Physician education and disease-specific knowledge are crucial to ensuring the appropriate information is communicated to the families in order to ensure linkage to specialty care for infants who screen positive [36]. 

Our study identified several barriers to the effective communication of the SCD diagnosis including PCPs’ lack of knowledge on how to interpret the NBS result and/or having the wrong PCP identified for the infant. These barriers are consistent with previous findings from a quality improvement project conducted in Wisconsin that also found that the communication of NBS results included inaccuracies of PCP listings provided by the birthing facility, high variability in PCPs’ descriptions of the results to the families, and difficulties in locating physicians willing to assume clinical responsibility for some carrier infants [37]. In another study, challenges associated with communicating results included situations in which parents changed PCPs (resulting in results sent to an incorrect location), families missing appointments when the results would have been discussed, missing NBS card, or incorrect infant information on the card (especially if a child’s name was not formalized at birth) [38]. This latter study also found that some parents did not understand the NBS results or had lingering concerns or questions that could not be answered by the PCP. These findings further emphasize the flaws in the NBS follow-up process [38]. 

It is crucial that families are informed of the SCD diagnosis by trusted, knowledgeable, empathetic providers. There are multiple factors that can influence a family’s experience, such as their current environment, previous family experiences, the content of the information delivered and the traits or behaviors of the person communicating the results. In our study, past family experiences with SCD were noted to be important. In particular, providers noted that families with previous experience with SCD had an easier time understanding the diagnosis; however, previous negative experiences with the healthcare system could preclude the acceptance of the diagnosis. This result is similar to findings from other studies where caregivers who have a child with SCD or a family history of SCD tend to more willingly accept the results. However, those families who have friends or community members with a child with severe SCD tend to have intense anxiety due to prior negative experiences with the disease [39]. In addition, parents who received results in a face-to-face setting as part of regular scheduled appointment reported that this method of disclosure was helpful [40]. These results are consistent with a qualitative study in the UK, where results received by letter caused more anxiety among parents compared to having a healthcare member personally disclose the result over the phone or in person [41]. As there is significant variation in how and where families are informed of the SCD diagnosis, additional work is needed to identify best practices to communicate these results. It is important to understand how this initial notification affects the child’s linkage to care or downstream, long-term care engagement. 

The communication and coordination of care for infants with suspected SCD ideally should include SCD providers, families, and SC-focused CBOs working in partnership to ensure optimal and timely linkage to care. Currently, this process is not standardized and improved collaboration and coordination is needed in addition to quality assurance. Several participants described attempts at centralizing data within their state using shared databases or data sharing agreements to improve communication. When feasible, these approaches were successful because they could identify infants who were not successfully linked to care. Similarly, a previous study from Germany in children with cystic fibrosis (another genetic condition) demonstrated the successful use of a centralized tracking mechanism for improving confirmatory testing and linkage to care [42]. A study in Brazil of NBS in SCD also highlighted efforts made by the Brazilian Ministry of Health to publish a national policy including guidelines for diagnosis, the communication of results, linkage to care, and the establishment of preventive care following a NBS diagnosis of SCD [43]. In the United States, NBS is conducted at the state level, which precludes the use of national policies; however, centralizing tracking at the state level that includes all relevant state-level stakeholders may be one mechanism for improving care.

Our study has important limitations and strengths to consider. First, our study did not include caregivers or primary care providers/pediatricians, so factors specific to communication and linkage to care from their perspective were not represented. We also included fewer states than previous studies; however, the high variability, even among these 11 states, was notable. Further, we had significant geographic representation in our population. Our previous work (although less in depth) included a larger sample of health care providers and similarly suggested high variability in practices [16]. Our study was strengthened by the inclusion of health care providers as well as DPH and CBO stakeholders, who provided important details and insight on processes specific to the NBS result up to establishing care with the SCD clinic. 

In conclusion, this qualitative study of NBS follow-up is the first part of the ENHANCE project to improve long-term engagement in care for children with SCD. This study found that more standardization is needed in how families are informed of a SCD diagnosis, ensuring that children undergo a confirmatory test and receive a referral to a SCD center. The communication of NBS results as well as referral and linkage to SCD specialty care are highly variable across states, precluding an assessment of “best practices” without a thoughtful prospective study. There are clearly many components to these processes that can be improved. Specific areas for future research include developing best practices for the disclosure of results, improving pediatrician knowledge and competence in reporting results, standardization in initial referral to care practices, and consistency in confirmatory testing and case closure. The development of robust tracking systems and data sharing may be particularly critical for facilitating these processes.

## Figures and Tables

**Figure 1 IJNS-10-00022-f001:**
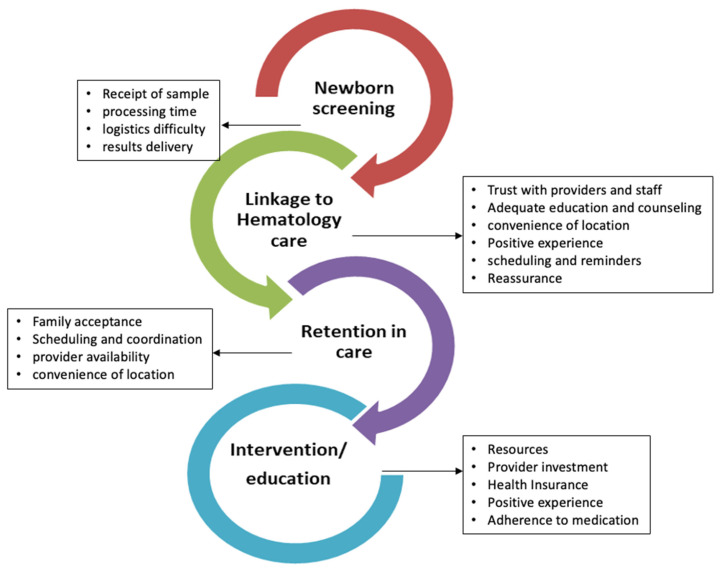
Sickle cell disease care continuum framework.

**Table 1 IJNS-10-00022-t001:** Demographic characteristics of interview participants: stakeholders (*n* = 21).

Variable	Mean, Median (Range), or *n* (%)
Gender	
Female	19 (90.5)
Male	2 (9.5)
Age (*n* = 4 prefer not to respond)	49.7, 50 (34–71)
Degree	
High School/Some College	2 (9.5)
Associates/Vocational	2 (9.5)
Bachelor’s	6 (28.6)
Master’s	9 (42.9)
Doctorate	2 (9.5)
Years experience at current NBS program	11.6, 8 (3–35)
Total years experience with NBS	13.6, 13 (3–35)

**Table 2 IJNS-10-00022-t002:** Demographic characteristics of interview participants: providers (*n* = 8).

Variable	Mean, Median (Range), or *n* (%)
Gender	
Female	7 (87.5)
Male	1 (12.5)
Age	47.8, 50 (31–58)
Provider type	
Registered Nurse	1 (12.5)
Nurse Practitioner	2 (25)
Medical Doctor	5 (62.5)
Years in clinical practice	19.3, 21.5 (8–26)
Years experience with SCD	17.7, 20 (5–34)
Specialty	
Pediatric hematology/oncology	5 (62.5)
Pediatric hematology nursing	1 (12.5)
Family nurse practitioner	1 (12.5)
Pediatric nurse practitioner	1 (12.5)
Patient age range	
0–21 years	4 (50)
0–25 years	4 (50)
Practice setting	
Urban	7 (87.5)
Mixed urban and rural	1 (12.5)

**Table 3 IJNS-10-00022-t003:** Themes and sub-themes with illustrative quotes.

Themes	Sub-Themes	Variability Between States	Illustrative Quotes
**NBS process and communication of results**	Determining initial diagnosis Communicating results to providers and families Barriers for communicating results to families. Facilitators for communicating results to families	*Communication of NBS results* Pediatricians notified Pediatricians and parents notified Parents notified directly CBO responsible for communicating to families *Barriers* Inconsistencies in contact information Families changing addresses Moving out of state Disorganized PCP practices Negative family responses-fear, emotional difficulty, etc. *Facilitators* Use of immunization database, Medicaid records, hospital records for tracking Use of case finders Dedicated and trained staff at the birth hospital to ensure correct contact information Partnerships with CBOs and sickle cell foundations	*“The state runs the test and then I’m notified via fax from the state. I will say in full disclosure it is a little bit disorganized”* *“Our health department…gets all the newborn screens, and if they’re positive for sickle disease, they notify the provider of record……The [department of health] also sends a letter to the family explaining the diagnosis, but the pediatrician is the one who actually notifies the family, ideally.”* *“I would say a big challenge is disorganization within practices. … Sometimes I speak to someone who forgets to tell the provider that I called, and I fax the information. Then the information gets lost or maybe they didn’t think it was important enough to tell their provider. … Sometimes we have providers that don’t fully understand what it is … I have definitely had a few situations where the provider told the family they had trait when it was [sickle cell disease], and they told the family that they had two types of trait, and that can be difficult and harmful, because sometimes the patients just stop trusting providers in general.”* *“If we can’t get in touch with the parents for some reason…then I’ll re-reach out to the primary and see what’s going on,….We have a great group of folks that’ll help. They’ll dig through Medicaid records, whatever they need to to try to find the parents and help them.”*
**Referral and linkage to SCD care**	Linkage to SCD care Barriers to linkage to SCD care Facilitators to linkage to SCD care Confirming SCD diagnosis Closing the case at the DPH	*Linkage to SCD care* Referred by the pediatrician Referred by the Health Department Family decision *Barriers* Gaps in communication between SCD clinic and DPH Families’ lack of acceptance of results Disorganized PCP practices System-level issues, complex systems for making appointments, misinterpretation of confirmatory diagnosis, lack of standards for reporting and registration *Facilitators* Good collaboration between SCD clinic, DPH, and PCPs Dedicated care coordinators Family support, education, patient advocates, financial assistance, and transport support *Confirmatory diagnosis* DPH responsible SCD clinic Pediatrician *Close out processes* First visit with the pediatrician First visit to the SCD clinic	*Yes, so the physician of record in the newborn period, so whoever gets the newborn screening results, so the primary care doctor or primary care provider, they get the newborn screening results from DHEC, and then the primary care provider refers the patient to us.”* *“Healthcare is not easy to access, and usually patients have to navigate through very complex phone tree systems, and it’s very difficult to get in touch with a human being, and very hard to get your phone call answered really, so there’s a lot of pitfalls along the way in that first appointment process.”* *Well, so in our state, we have State Children’s Services. The nice thing is for every positive, whether it’s a sickle cell or thalassemia positive,…. the state children’s service gives an approval, its an expedited approval that says this patient is to be seen as soon as possible and they will pay for that visit. Regardless of insurance, regardless of coverage.”* *“Ideally, we’d get them in within 30 days…Per the state, you really would like to see them get confirmatory testing within 90 days. I would say for the most part we’re getting confirmatory testing between day 15 and day 90.”* *“When I go to close the case on Newborn Surveillance Tracking and Reporting, I copy the follow up into the …system, and I check [EMR] to make sure I get all my dates right. I close it out from the state system …which usually happens after the baby’s first appointment at [SCD clinic] typically. Obviously, there are some exceptions to that, like if we weren’t able to get labs at the first appointment, but for the most part, I close it out after that first appointment.”*

Abbreviations: DPH—Department of Public Health; SCD—sickle cell disease; NBS—newborn screening; CBO—community-based organization.

## Data Availability

The data presented in this study are available on request from the corresponding author.

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
