# Peer review of "Current Methods of Newborn Screening Follow-Up for Sickle Cell Disease Are Highly Variable and without Quality Assurance: Results from the ENHANCE Study"

_2409-515X, 2024, doi:10.3390/ijns10010022_

Round 1
Reviewer 1 Report
Comments and Suggestions for Authors
This manuscript describes a qualitative study of newborn creening practices for sickle cell disease across a selected subset of states in the US. It is focused onot on the laboratory testing but on what comes next in the newborn screening process- communication of results and initiation of approprate care. It identifies significant variation in practices across states. While varaible practices are not necessarily a bad thing and different practices were not correlated specifically with outcome, it does highlight some areas of focus for further investigation and potential improvement. It is well written and clear with approprate references.
Author Response
Thank you for taking time to review our manuscript.
Reviewer 2 Report
Comments and Suggestions for Authors
This article evaluates current methods of newborn screening follow-up for sickle cell disease. Follow-up after a positive screening result is an important issue. However, in my opinion, the paper could be improved by focusing on the main points and avoiding repetition. In particular, the results are redundant and much too long.
In general: Please make sure that all abbreviations are explained (i. e. NHLBI line 77; HIPAA line 122; RN, NP, MD table 1; PCP table 2.
Abstract:
The abstract should state the important facts of the study and not summarize the introduction or describe the detailed results. Therefore, I would suggest deleting line 15-20 and 27-30. Line 33: facilitators
Introduction:
ENHANCE is only in the title, maybe you can describe the study
Line 39: why only the prevalence for black people and not in general (SCD also affects i. e. other sub-Saharan and Asian ethnicities)?
Line 54, 62 and 77: References should be included
Methods: In the results 29 interviews were done, here only 27? (line 98)
Results:
Table1: Are race, Ethnicity and patient age range important for the results? Otherwise delete
The important and interesting part of the results is table 2-here all the main results for the different themes, barriers, facilitators and illustrative quotes are perfectly summarized. The written text often repeats these results from table 2 and gives little additional information. So please shorten this chapter and focus on new important points to improve the paper. For example, in my opinion, in 3.2. line 173-180 and 183-190 can be deleted. The illustrative quotes should not be repeated in the text (line204-207, 223-228, 254-259…..).
Discussion:
Please don’t repeat the detailed results (i. e. line 412-416, 463-474).
Are there programs for follow-up in other countries or for other diseases that can be discussed? Was there a difference in the answers of stakeholders and providers? Was there an impact on responses by race/ethnicity? Was any state overrepresented?
Perhaps it would be interesting to additionally discuss wether follow-up is better with some of the described communication strategies or linkages to SCD care? Maybe the insufficient NBS follow-up (30% penicillin prophylaxes, no stroke screening) is because the children are lost to follow-up or is the treatment in the centers inadequate?
Comments on the Quality of English LanguageMinor editing of English language required
Author Response
Thank you for taking time to review the manuscript. Please find attached the detailed responses and the corresponding revisions/corrections highlighted in track changes in the re-submitted files. We have also attached a clean copy of the revised manuscript .

Round 2
Reviewer 2 Report
Comments and Suggestions for Authors
The paper is much better now, all issues have been adressed